# Astrocytic Connexin43 Channels as Candidate Targets in Epilepsy Treatment

**DOI:** 10.3390/biom10111578

**Published:** 2020-11-20

**Authors:** Laura Walrave, Mathieu Vinken, Luc Leybaert, Ilse Smolders

**Affiliations:** 1Department of Pharmaceutical Chemistry, Drug Analysis and Drug Information, Research Group Experimental Pharmacology (EFAR), Center for Neurosciences (C4N), Vrije Universiteit Brussel, Laarbeeklaan 103, 1090 Brussels, Belgium; Laura.Walrave@VUB.be; 2Department of In Vitro Toxicology and Dermato-Cosmetology, Vrije Universiteit Brussel, Laarbeeklaan 103, 1090 Brussels, Belgium; Mathieu.Vinken@VUB.be; 3Physiology Group, Department of Basic and Applied Medical Sciences, Ghent University, C. Heymanslaan 10, 9000 Ghent, Belgium; Luc.Leybaert@UGent.be

**Keywords:** connexin43, seizures, epilepsy, gap junctions, hemichannels, astrocytes, mimetic peptides

## Abstract

In epilepsy research, emphasis is put on exploring non-neuronal targets such as astrocytic proteins, since many patients remain pharmacoresistant to current treatments, which almost all target neuronal mechanisms. This paper reviews available data on astrocytic connexin43 (Cx43) signaling in seizures and epilepsy. Cx43 is a widely expressed transmembrane protein and the constituent of gap junctions (GJs) and hemichannels (HCs), allowing intercellular and extracellular communication, respectively. A plethora of research papers show altered Cx43 mRNA levels, protein expression, phosphorylation state, distribution and/or functional coupling in human epileptic tissue and experimental models. Human Cx43 mutations are linked to seizures as well, as 30% of patients with oculodentodigital dysplasia (ODDD), a rare genetic condition caused by mutations in the GJA1 gene coding for Cx43 protein, exhibit neurological symptoms including seizures. Cx30/Cx43 double knock-out mice show increased susceptibility to evoked epileptiform events in brain slices due to impaired GJ-mediated redistribution of K^+^ and glutamate and display a higher frequency of spontaneous generalized chronic seizures in an epilepsy model. Contradictory, Cx30/Cx43 GJs can traffic nutrients to high-energy demanding neurons and initiate astrocytic Ca^2+^ waves and hyper synchronization, thereby supporting proconvulsant effects. The general connexin channel blocker carbenoxolone and blockers from the fenamate family diminish epileptiform activity in vitro and improve seizure outcome in vivo. In addition, interventions with more selective peptide inhibitors of HCs display anticonvulsant actions. To conclude, further studies aiming to disentangle distinct roles of HCs and GJs are necessary and tools specifically targeting Cx43 HCs may facilitate the search for novel epilepsy treatments.

## 1. Introduction

Despite the availability of various anti-epileptic drugs (AEDs), still a significant number of patients with epilepsy (estimated to be 30–40%) remain refractory to pharmacological treatments [1]. Although refractoriness to AEDs is likely multifactorial, a crucial element is that most AEDs have similar mechanisms of action, which is mainly targeting neuronal voltage-gated ion channels and facilitation of gamma-aminobutyric acid (GABA)ergic inhibition [2]. Exploring new glial targets can thus offer therapeutic gain for the treatment of refractory seizures. Dysfunctional astrocytes are indeed recognized as critical contributors to epileptic activity [3,4]. Astrocytes are the most prominent cell type in the brain and harbor several cell-type-specific proteins, of which their expression is altered in epilepsy [5]. Briefly, malfunctioning of astrocytic excitatory amino acid transporters (EAATs) type 1 and 2 [6], glutamate-converting enzyme glutamine synthetase (GS) [7], adenosine-converting enzyme adenosine kinase [8], spatial K^+^ buffering by inward rectifying K^+^ (K_ir_) channels, and connexin (Cx)-based gap junctions (GJs) and water regulation by aquaporin (AQP) channels [9], all influence neuronal excitability and change the cellular environment to a seizure-prone condition with increased excitation/decreased inhibition [3,4]. In addition, astrocytic connexin (Cx) hemichannels (HCs) seem to add to the pathophysiology of epilepsy [10,11,12,13,14]. This paper focuses on Cx43, since it is the most abundant and widespread astrocytic Cx throughout the brain and hippocampus [15], a brain region crucially involved in epilepsy. 

Cxs are tetraspan proteins containing four transmembrane (TM) domains, two extracellular loops (ELs), one cytoplasmic loop (CL), and a C- and N-terminal tail (CT and NT), and are the constituents of two channel types, GJs and HCs [16]. They are represented by 21 different isotypes in the human genome and 20 isotypes in the mouse genome [17,18,19], of which 11 expressed in the brain [20]. In the central nervous system (CNS), Cx43 is expressed in developing neurons, activated microglia, pericytes, and endothelial cells of blood vessels, but is predominantly found in astrocytes [21,22,23,24,25]. Astrocytic Cx43 HCs are composed of six Cx43 subunits and are located between the cytosol of an astrocyte and its extracellular environment [16,26]. They are usually closed in physiological conditions and may open in pathological conditions. Cx43 HCs have a single channel conductance of ~220 pS and open at low extracellular Ca^2+^ concentration ([Ca^2+^]_e_) or moderately increased intracellular Ca^2+^ concentration ([Ca^2+^]_i_) [27,28,29,30], extracellular alkaline pH [31], or by cytokines (e.g., tumor necrosis factor α (TNF-α) and interleukin 1-beta (IL-1β)) [32], oxidative stress [33], dephosphorylation in ischemia [34], mechanical stimulation [35,36,37] and others. Noteworthy, the CT is known to be a pH sensor and is important in channel gating [38,39]. As such, Cx43 HCs are often seen as pathological pores, although a potential role of Cx43 HCs in cognitive functions, synaptic transmission, and plasticity have recently been demonstrated, suggesting that Cx43 HCs may open under basal, physiological conditions as well [40]. On the other hand, Cx43 GJs are formed by the docking of two Cx HCs of adjacent cells [16,26] and are endowed with important physiological functions in the context of the astrocytic syncytium, facilitating the distribution of glucose and lactate [41,42], the spatial buffering of K^+^ and glutamate [43,44,45], and maintaining isopotentiality of the astrocytic network [46]. They close under specific conditions such as a low intracellular pH (<6.5), high [Ca^2+^]_i_ [28] or strong transjunctional voltage differences, in order to prevent the transfer of death signals [26,47]. Noteworthy, intramolecular CT-CL interactions are important for Cx43 channel gating; in particular, interactions between the CT and the CL inhibit Cx43 GJs, whereas they facilitate Cx43 HC opening in response to chemical stimulation [48].

Cx43 GJs have a complex and ambiguous role in epilepsy and two (divergent) possibilities emanate from literature [49] (Figure 1). Gap junctional intercellular communication (GJIC) is believed to be key in maintaining tissue homeostasis and in propagating electric and metabolic signals across cell populations [16]. In line with this, GJs could suppress seizure activity by redistributing K^+^ (which otherwise rises to ictogenic levels) [9] and glutamate [50,51], away from areas with high activity. However, K^+^ dynamics in epilepsy are complex as high K^+^ is able to abort seizures as well [52] when its concentration rises to levels that inactivate Na^+^ channels. On the other hand, Cx43 GJs might promote seizure generation by enhancing glial synchronization (Ca^2+^ wave propagation) during seizures [5]. Indeed, although astrocytes are electrically non-excitable, they should be considered as chemically excitable through their Ca^2+^ signaling machinery [53]. Increases in astrocytic [Ca^2+^]_i_ and propagating Ca^2+^-waves between astrocytes can activate Ca^2+^-dependent ion channels and induce glutamate release from astrocytes, which contributes to seizure generation [4], neuronal synchronization, and spread of ictal activity [54]. Furthermore, GJs facilitate nutrient distribution in the astrocytic syncytium and are involved in the (indirect) supply of nutrients to neurons in an activity-dependent way [42]. In epilepsy, they could therefore help to cover the high energy demand generated during seizures [41,42]. The biggest challenge however in unravelling the nature of the involvement of the GJ-coupled astrocytic network in epilepsy, is the lack of Cx-isotype specific inhibitors, invariably resulting in inhibition of neuronal Cx36.

Interestingly, Cx43 HCs, are frequently considered as pathological pores, because uncontrolled HC opening can lead to cellular damage via several mechanisms including (i) HC Ca^2+^ entry and elevation of [Ca^2+^]_i_, (ii) Na^+^ and Cl^−^ entry and cell swelling, and (iii) spread of toxic molecules released by HCs (e.g., glutamate) from injured cells [55]. Additionally, Cx43 HCs participate in astrocytic Ca^2+^ wave propagation (and glial synchronization) by releasing adenosine triphosphate (ATP) in the extracellular space [56], and in the activation of neuronal N-methyl-D-aspartate (NMDA) receptors by the release of gliotransmitters (glutamate and D-serine; the latter is controversial as the racemase has been claimed to be present in neurons, not astrocytes [57]) [58]. Noteworthy, astrocytic gliotransmitter release is mainly mediated by elevations in astrocytic [Ca^2+^]_i_ [59], but mechanisms that have been proposed include vesicular as well as non-vesicular mechanisms including passive diffusion via Cx HCs. While previous studies have focused on the role of Cx43 GJs in seizures, little evidence exists for the involvement of Cx43 HCs mainly due to the lack of selective pharmacological inhibitors that can differentiate GJ versus HC functions. Moreover, Cx43 HCs do not solely function as pathological pores but are also suggested to play roles in higher brain functions as well [55]. In the following sections, we review the available data specifically on glial Cx43 HC and GJ involvement in seizures and epilepsy and discuss the potential of Cx43 channels as targets for epilepsy. Reviews on multiple types of neuronal/glial Cx channels in epilepsy can be found in [52,60,61,62,63].

## 2. Altered Cx43 Expression, Phosphorylation State, Coupling and Distribution in Human Tissue from Patients with Epilepsy

### 2.1. Cx43 mRNA and Protein Expression Studies

Studies of cerebral tissue from patients with epilepsy of mainly structural etiology show upregulation of glial, but not neuronal Cxs, favoring the idea that glial Cxs are involved in seizures [64]. Several research groups provide evidence of Cx43 involvement in human epilepsy (Table 1) and mRNA and protein expression analyses from tissue excised from patients with epilepsy of structural etiology seem quite consistent, with most studies reporting increased expression of Cx43 in epileptic tissue. Naus and coworkers first reported increased Cx43 mRNA expression in temporal lobe neocortex from patients with intractable seizures and in neocortex from cerebral tumors that give rise to acute seizures [65]. Augmented Cx43 protein levels were confirmed by other research groups (in epilepsy associated brain tumors [66,67], in (mesial) temporal lobe epilepsy ((M)TLE) [67,68,69,70], in epilepsy secondary to focal cortical dysplasia (FCD) type IIB [71], and in epileptic foci of refractory seizures [72]). In contrast, decreased Cx43 mRNA and unaltered protein levels were also reported in hippocampal tissue from patients with a complex partial seizure disorder [73].

However, interpreting human studies can be complicated since epilepsy is a very heterogeneous disorder and other factors can impede the interpretation. First, Cx43 expression analysis was always performed in tissue resected from patients with medically intractable seizures, often from patients with (M)TLE and thus epilepsy of structural etiology. As opposed to tissue obtained from experimentally induced animal models, this tissue has limited availability, and therefore has only restrictedly been examined. Variations can originate due to differences in type of epilepsy, age of patients, seizure duration, and stage of the condition [62]. Besides this, patients frequently take several AEDs, which in turn can alter Cx expression. A recent study by Fukuyama et al. demonstrated the distinct effects of three voltage-dependent Na^+^ channel-inhibiting AEDs on the astroglial release of glutamate and ATP in vitro using primary cultured astrocytes. Zonisamide subchronically decreased Cx43 expression and acutely/subchronically inhibited Cx43 HC activity. Lacosamide acutely inhibited HC activity but did not subchronically affect Cx43 expression. Therapeutic-relevant concentrations of carbamazepine did not affect HC activity or Cx43 expression, while a supratherapeutic concentration decreased Cx43 expression and HC activity [75]. In line with this, they demonstrated that in a genetic model of autosomal dominant sleep-related hypermotor epilepsy (ADSHE), zonisamide chronically reduced thalamic Cx43 expression, and acutely suppressed thalamocortical glutamatergic transmission via inhibition of thalamic HC activity, whereas subchronic and acute administration of carbamazepine did not affect thalamic Cx43 expression and HC activity [76]. Some common AEDs including carbamazepine, gabapentin, and valproic acid (24 h incubation) did not alter astrocytic Cx43 expression in astroglia/microglia co-cultures [77], although this does not exclude possible adaptations in patients following chronic AED treatment. Levetiracetam reversed the downregulation of Cx43 expression and impaired GJIC in astrocytes co-cultured with 30% microglia (i.e., mimicking inflammation) [78]. Moreover, TLE is often associated with massive gliosis and activated astrocytes, which also changes Cx43 expression and function [62]. In human studies, comparisons are made with control tissue arising from autopsy specimens (e.g., dissected from sudden death patients) and brain tumors. In the latter case, Cx expression can also be modified [79]. Accordingly, Aronica et al. found increased Cx43 protein levels in low-grade gliomas and perilesional cortex, and decreased protein expression in high-grade gliomas, implicating Cx43 involvement in tumor-related seizures [66]. Noteworthy, it is still unclear whether changes in Cx43 expression have any direct effect on seizure generation [62].

### 2.2. Cx43 Coupling, Phosphorylation, and Distribution Studies

Changes in mRNA and protein levels do not necessarily translate into changes in functional coupling. As hypothesized by Elisevich et al., ictogenicity may be related to a change in the dynamic properties of the existing channels (open versus closed) rather than an upregulation in the absolute number of channels [73]. Few functional data on GJIC are available in human tissue and further studies using fluorescence recovery after photobleaching (FRAP) analysis of (fresh) ex vivo human tissue slices would help to better understand the role of GJIC in epilepsy [79]. Lee et al. used the FRAP technique to quantify GJIC from tissue surgically resected from patients with medically intractable epilepsy (structural etiology) and found that GJIC was more pronounced in astrocytes isolated (and cultured) from hyperexcitable tissue, but primary astrocyte cultures used were 2–3 weeks old [80], making it rather unlikely that these cells maintain functional properties as in the diseased tissue [41]. Controversial results in expression studies may also be partly due to the fact that Cx43 proteins are tightly regulated by post-translational modifications that can influence protein oligomerization, trafficking, membrane insertion and aggregation, GJIC and open probability, unitary conductance and internalization from the membrane [74,81]. This includes ubiquitination, phosphorylation, S-nitrosylation, which can affect Cx43 channel opening/closing dramatically [81]. More specifically, phosphorylation acts as a major gatekeeper in the normal Cx43 lifecycle. Cx43 is primarily phosphorylated at serine residues and exists in multiple isoforms after electrophoretic separation, namely the non-phosphorylated (P0~41 kDa) or phosphorylated (P1~44 and P2~46 kDa) variants [82]. Recent reports utilizing phosphospecific antibodies have shown that phosphorylation at S365 is necessary for the shift to the P1 isoform [83]. This P1 isoform was found primarily in the plasma membrane and in a subset of GJ plaques. Similarly, phosphorylation at S325/328/330 is involved in the shift to P2 and this isoform was found exclusively in GJs [84].

Ex vivo hippocampal tissue of MTLE with hippocampal sclerosis (HS) was completely devoid of normal astrocytes and GJIC, whereas coupled astrocytes were abundant in non-sclerotic MTLE tissue [85]. Deshpande et al. nicely demonstrated that this loss of GJIC is due to redistribution of Cx43 protein and/or altered phosphorylation in CT sites, affecting channel permeability [74]. Cx43 membrane protein expression was unaltered in HS versus non-HS tissue, but Cx43 accumulated in large plaques at presumed end feet around blood vessels. Phosphorylation at S255 was increased, i.e., a site known to reduce open probability of GJs upon phosphorylation. The Cx43 dysregulation resulted from blood–brain barrier (BBB) damage and consequential albumin extravasation in HS, which was absent in non-HS tissue. Extravasation of albumin, present in hippocampus of patients in the chronic phase of TLE, leads to transforming growth factor beta (TGF-β)-mediated signaling cascades [86] and involves activation of mitogen-activated protein kinases (MAPKs). MAPK can initiate transcription of several proinflammatory cytokines, or directly affect Cx43 phosphorylation [74]. In line with this, Braganza et al. reported that in mice, intracerebroventricularly (i.c.v.) injected albumin is taken up by astrocytes and entails a significant decrease (33%) in astrocytic GJIC in hippocampus one day post-injection [87]. Interestingly, MAPK phosphorylation of Cx43-based S255, S262, S279, and S282 is necessary for astrocytic Cx43 HC opening [88].

## 3. Link between Human Inherited Cx43 Mutations and Seizures

Oculodentodigital dysplasia (ODDD) is a rare genetic disease caused by mutations in the GJA1 gene located on chromosome 6, encoding for Cx43. In most cases, it is inherited as autosomal dominant (Online Mendelian Inheritance in Man (OMIM) #164200), but cases of autosomal recessive inheritance (OMIM #257850) are also reported. There are 62 known GJA1 mutations (mostly missense mutations), found in all protein domains, that lead to Cx43 protein alterations (e.g., altered channel oligomerization, trafficking, gating, and permeability) [89,90]. Since Cx43 is ubiquitously present in the human body, multiple tissues and cells are affected. ODDD patients therefore exhibit a pleiotropic phenotype with craniofacial, ocular, dental, and limb anomalies [89] that are sometimes associated with neurological problems [91]. Since astroglial Cx43 channels actively contribute to BBB integrity, nutrient supply to neurons [92], release of gliotransmitters, propagation of intracellular Ca^2+^ waves, spatial K^+^, and glutamate buffering, it is not surprising that around 30% of ODDD patients display neurological symptoms [89], including seizures [93].

Dobrowolski et al. characterized four Cx43 mutations (I31M, G138R, G143S, and H194P) after stable expression in HeLa cells and found that these mutations led to lack of the P2 phosphorylation state of the Cx43 protein (important for GJ function; all mutations), to complete inhibition of GJIC (all mutations) and to increased HC activity (I13M, G138R, and G143S mutations). They hypothesized that increased Cx43 HC activity may aggravate the phenotypic abnormalities in ODDD patients who are deficient in Cx43 GJ channels [94]. Although further research is required, the epileptic phenotype in ODDD patients supports the hypothesis that Cx43 is involved in seizures.

## 4. Altered Cx43 Expression, Phosphorylation State, Coupling and Distribution in Rodent Models of Seizures and Epilepsy

### 4.1. Cx43 mRNA and Protein Expression Studies

Several studies show changes in Cx43 mRNA and/or protein expression in rodent models of seizures, status epilepticus (SE), and epilepsy in the hippocampus (Table 2) and other seizure-relevant regions (Table 3). Both increased [64,76,95,96,97,98,99,100,101,102,103,104,105,106,107], decreased [108,109,110,111,112,113], as well as unaffected [74,114,115,116,117,118] Cx43 expression is described, and results can be contradictory, impeding the ability to draw definitive conclusions. Factors leading to variations can be the species and age of animals, intrinsic differences of the method to induce seizures (e.g., chemically versus electrically induced seizures), the duration of seizure activity, seizure frequency, and the brain region examined in each study [61]. Additionally, protein expression may be differentially regulated over the course of epilepsy (i.e., acute versus chronic stage) within a certain animal model. For instance, Takahashi et al. and Söhl et al. both used the systemic (intraperitoneal, i.p.) kainic acid SE rat model and found respectively increased Cx43 protein levels in the latent phase 7 days after SE [104] and unaltered Cx43 expression in the chronic phase 4 weeks after SE [117]. While Cx43 expression studies are important, other more functionally-linked parameters are necessary to grasp the whole picture of seizure/epilepsy induced Cx43 alterations; two such parameters are post-translational modifications and functional coupling.

### 4.2. Cx43 Phosphorylation, Coupling and Distribution Studies

Khan et al. revealed that the fractional contribution of the phosphorylated isoform 1 (P1)/ phosphorylated isoform 2 (P2) bands to total protein was significantly higher in juvenile mice with hyperthermia (HT)-induced febrile seizures [120]. Significantly increased P1, and slightly increased P2 and P0 Cx43 was reported in Co^2+^-induced epileptiform activity in hippocampal slices [64]. Deshpande et al. harvested hippocampi 3 months after intracortical kainic acid injection (MTLE model) and found unaltered Cx43 plasma membrane protein levels, but a significant shift towards the P2 band, indicating altered Cx43 phosphorylation. Phosphorylation was increased at sites S255 and S368, i.e., two sites that entail reduced open probability (S255) and a smaller unitary conductance (S368) of Cx43 GJ channels, which might contribute to loss of astrocyte coupling [74]. Cx43 phosphorylation will however also influence HCs [121], but currently the available information is limited. In the hippocampus, increased astrocytic coupling is reported ex vivo after bicuculline (GABA_A_ receptor antagonist)-induced epileptiform activity in hippocampal slices [101] and in vivo in the post-SE kainic acid model [104]. By contrast, Xu et al. and Khan et al. demonstrated reduced coupling in vivo in a model of tuberous sclerosis complex (a genetic benign brain tumor disease accompanied by seizures) [112], and of experimental febrile seizures [120]. Since astrocyte coupling in the hippocampus is mainly accomplished by Cx43 and only to a minor extent by Cx30 [15], these results suggest altered Cx43 GJIC in experimental seizure models. 

An additional factor, not reported by mRNA and/or protein expression studies, concerns the cellular distribution of Cx43. A pronounced cellular redistribution of Cx43 around blood vessels was found in the sclerotic mouse hippocampus, but further investigations are necessary to determine the functional consequences thereof [74]. Two hours after onset of pilocarpine-induced SE in rats, hippocampal Cx43 immunolabeling was increased in cornu ammonis 3 (CA3) stratum pyramidale (171%) but reduced (48%) in the stratum radiatum [122]. The authors hypothesized that the increase in the stratum pyramidale layer is due to the high energy demand promoted by sustained epileptic activity and the increase could account for two opposite situations. Indeed, increased Cx43 GJIC may provide spatial buffering of K^+^ and neurotransmitters and facilitate HC ATP release that is degraded to adenosine which acts neuroprotectively; alternatively increased GJIC may enhance propagation of toxic metabolites and death signals (e.g., reactive oxygen species, glutamate, ATP [123]) to adjacent astrocytes, while HCs may contribute in dispersing these substances in a paracrine way. Decreased Cx43 labeling in stratum radiatum is unlikely to have great impact on spatial K^+^ buffering during SE, since in this layer, astrocytes have a more elongated morphology, thereby reducing their need for GJ coupling in K^+^ redistribution [124] (further discussed below). 

## 5. Data Derived from Transgenic Mice

Theis et al. described an astrocyte-targeted deletion of Cx43 in adult mice, which severely reduces inter-astrocytic cell coupling [125]. However, coupling was not completely abolished due to compensatory changes of Cx30 [124,125,126]. To avoid this, Wallraff et al. obtained mice with complete coupling-deficient astrocytes by crossing conditional Cx43-deficient mice with total Cx30 knock-out (KO) (Cx30^−/−^) mice [127] (i.e., homozygous double-deficient Cx30^−/−^, Cx43^fl/fl^:hGFAP-Cre (double KO (dKO) mice)) [124]. Astroglial Ca^2+^ waves were completely abolished in Cx30/Cx43 dKO mice, while in Cx30 KO mice with functional Cx43, no alterations in astrocytic Ca^2+^ waves were observed. The fundamental morphological and functional properties of the astrocytes were unchanged. However, the mice showed prominent white matter abnormalities due to loss of astrocyte-oligodendrocyte coupling, similar to those noted in Cx32-Cx47 dKO mice [124]. In the hippocampus, a surprisingly large capacity for K^+^ clearance was conserved so Cx30/Cx43 GJ-dependent processes only partially account for K^+^ buffering in the hippocampus. In fact, this depended on the molecular layer studied: in stratum radiatum layer (~300 µm away from stratum pyramidale), radial K^+^ relocation was unaltered (so not dependent on Cx GJIC), while in stratum lacunosum molecular layer (~400–500 µm away from stratum pyramidale), K^+^ relocation was impaired. This different result can be explained by the organization of astrocytes in each layer. In stratum radiatum, single astrocytes have a bipolar shape and are perpendiculary orientated to the stratum pyramidale. A single astrocyte spans much larger areas than in stratum lancunosum moleculare, where astrocytes are in fact round and have no particular orientation. The perpendicular array of individual astrocytes in stratum radiatum makes these cells ideally suited for spatial buffering of K^+^ released by pyramidal cells, independent of GJs. The partial overlap of elongated astrocytic processes might facilitate the release of K^+^ from one astrocytic pole and uptake at the next via K^+^ channels (i.e., via indirect coupling) [124]. Nevertheless, Cx30/Cx43 dKO slices showed increased susceptibility for generation of epileptiform activity. Six out of 10 slices developed spontaneous epileptiform activity and only slices of dKO mice generated epileptiform activity during low-intensity Schäffer collateral stimulation, compared to wild type slices. Since K^+^ buffering may become particularly important under conditions of synchronized, high-frequency neuronal firing, such as during seizure-like discharges, slices were exposed to zero extracellular Mg^2+^. Spontaneous seizure-like discharges induced by Mg^2+^-free solution recorded in the CA1 stratum pyramidale occurred with shorter latency and higher frequency in dKO slices [124]. Cx30/Cx43 dKO mice were subjected to an in vivo kainic acid MTLE model and, despite having a similar severity of SE as their wild type littermates, displayed a higher frequency of spontaneous generalized chronic seizures, supporting the view that loss of astrocyte coupling represents a crucial event in epileptogenesis [128].

Astrocytes from Cx30/Cx43 dKO mice still manage to take up high levels of released K^+^ and glutamate, although they are unable to redistribute them via the panglial network [45]. Intracellular accumulation of high amounts of K^+^, glutamate and osmotically attracted water leads to astrocytic cell swelling, thereby significantly reducing the extracellular space volume [45]. This can contribute to increased neuronal excitability by augmenting extracellular concentrations of ions and neurotransmitters. Elevation of extracellular K^+^ will result in astrocyte depolarization that will counteract the electrogenic glutamate transporter 1 (GLT-1; i.e., rodent EAAT2) activity, impeding glutamate clearance [129].

Rouach et al. revealed pro-epileptic effects of astroglial Cx43 and Cx30 GJs, as they provide energetic metabolites from blood vessels to neurons in an activity-dependent manner [42]. Indeed, perivascular astrocytes can take up glucose from the blood by the glucose transporter 1 (GLUT-1) located in their end feet as well as in BBB endothelial cells. Subsequently, glucose is metabolized into lactacte, which can traffic through astroglial GJs and is used by neurons as energy substrate to sustain their excitatory synaptic activity [42]. They found that this intercellular trafficking of glucose and lactate is regulated by glutamatergic synaptic activity mediated via α-amino-3-hydroxy-5-methyl-4-isoxazolepropionic acid (AMPA) receptors. Increased glutamate did not appear to increase GJ permeability but rather acted by an increased energetic demand that generates a diffusion gradient for glucose [42].

In conclusion, these studies show that the role of astroglial Cx43 GJIC in the pathophysiology of epilepsy is still controversial. GJIC may act in an anticonvulsant manner by facilitating spatial K^+^ and glutamate buffering, and extracellular space volume regulation. On the other hand, GJIC could promote ictogenesis by mediating nutrient transport to neurons to sustain synaptic activity and by facilitating the synchronization of neuronal electrical activity via astrocytic Ca^2+^ astrocytic signaling [49]. One should be cautious to draw conclusions from KO studies, since in dKO mice, Cx30 and Cx43 are permanently deleted, HC and GJ channel functions are eliminated, non-channel functions also disappear, and other compensatory mechanisms are possible [125,126,130,131]. These compensatory mechanisms include increased GLT-1 (i.e., rodent EAAT2) and glutamate aspartate transporter (GLAST; i.e., rodent EAAT1) protein levels in the cerebral cortex but not in hippocampi [126].

Cx43-G138R transgenic mice with an astrocyte-targeted point mutation of Cx43 (glycine 138 substituted with arginine) represent a model of hereditary ODDD and have deficient GJIC but increased HC function as judged from ATP release [132]. To the best of our knowledge, these mice have not yet been subjected to seizure models, however were shown to have enhanced astroglial Ca^2+^ waves [56]. In particular, the slow (astrocyte-linked) Ca^2+^ wave propagated ~40% further in Cx43G138R slices compared to littermate controls, and the Ca^2+^ increase was prolonged. Given the fact that Cx43 HC ATP release may act as a trigger of Ca^2+^ waves, these channels might possess proconvulsive effects. The ambiguous roles of glial Cx HCs/GJs in epilepsy reveal an urgent need to investigate the acute effects of Cx43 channel inhibition by using pharmacological agents of Cx channels.

## 6. Effects of General Cx Channel Blockers/Openers in Rodent Seizures and Epilepsy Models

### 6.1. In Vitro Seizure Models

Another approach to assess the role of Cx43 channels in epilepsy is their pharmacological inhibition. In this section, we discuss general Cx channel blockers that inhibit both GJs and HCs. In many cases, these inhibitors also affect other targets and care should be taken in the interpretation of results obtained with these substances. Carbenoxolone (CBX) is a widely-used GJ/HC blocker that rapidly and reversibly blocks GJs and HCs in cell cultures and acute brain slices [133]. CBX is a glycyrrhetinic acid derivate that lowers plasma membrane fluidity and acts via various pathways including protein kinases, G-proteins, transport ATPases, and Cx phosphorylation to inhibit junctional conductance [61]. CBX is water-soluble and has poor BBB permeability [134] and has toxic effects on mitochondria [135]. It does not discriminate between Cx isotypes and has a large spectrum of effects, including inhibition of Cx-based GJs and HCs (astrocytic Cx43/Cx30 [136,137], neuronal Cx36 [138]), as well as pannexin (Panx) channels) [139]. Moreover, other off-target effects of CBX, such as inhibition of voltage-gated Ca^2+^ channels [140], P2X purinoreceptor 7 [141], 11β-hydroxysteroid dehydrogenase [142], and GABA and AMPA receptor-mediated synaptic transmission [143,144], all largely affect excitability [140,145] and complicate the interpretation of pharmacological studies with this compound. In epilepsy research, CBX was reported to have antiepileptiform effects in dozens of in vitro seizure models [14,101,115,146,147,148,149,150,151,152,153,154] with only one in vitro study showing opposite effects [155] (Table 4). 

Attempts to rule out off-target effects of CBX, as for instance done by Kékesi et al. and Vincze et al. by comparing in vitro results with an antibody directed against the Cx43 C-terminal end [147,154], are generally lacking in literature. Of note, the antibody in this study was extracellularly applied, making it unclear how a ~150 kDa protein could make its way into the cell and bind to the Cx43 C-terminal located target; additionally, if the antibody was able to enter the cell, the outcome of interfering with Cx43 channel function is not established and was not tested. 

As an alternative to CBX, compounds of the fenamate family are reported to be potent blockers of Cx43-mediated GJIC with the following order of efficacy: meclofenamic acid > niflumic acid > flufenamic acid [158]. The effect does not involve changes in intracellular Ca^2+^ or pH and is unrelated to protein kinase C activity or inhibition of cyclo-oxygenase activity [61]. Flufenamic acid suppresses epileptiform activity in hippocampal slices induced by 4-aminopyridine (voltage-gated K^+^ channel blocker) and zero extracellular Mg^2+^ [159], and eliminates seizure-like events in bicuculline-treated neocortical slices [160]. However, fenamates are used as anti-inflammatory drugs and analgesics, and besides being non-selective towards other Cx isotypes, they also have effects on voltage-gated K^+^ channels and GABA receptors [161,162]. Considering the role of inflammation in epilepsy, the question remains whether fenamates affect seizures via Cx channel inhibition or by inhibiting anti-inflammatory cascades [163]. Most interestingly, it was recently shown that prenatal lipopolysaccharide exposure enhances Cx43 HCs in the hippocampus of adult offspring mice, affecting glutamate release, [Ca^2+^]_i_ handling, astrocyte morphology, and neuronal function [164]. 

Intracellular alkalinization enhances GJ coupling and increases epileptiform activity. In rat hippocampal slices with primary discharges induced by Mg^2+^-free solution, trimethylamine (a weak base, inducing transient intracellular alkalosis) induced secondary (i.e., lasting several hundred milliseconds) and tertiary (i.e., lasting for several seconds) bursts [148] and decreased the onset and interictal interval of seizure-like events [154]. In addition, ammonium chloride increased frequency and duration of burst activity [156] and evoked spontaneous secondary afterdischarges following repetitive Schäffer collateral tetanization in rat hippocampal slices [115]. However, it must be remembered that intracellular alkalinization does not only affect astrocytic Cx43 channels but also neuronal and vascular Cxs, as well as many other ion channels and processes [61].

To conclude, all the above-mentioned drugs tend to have only partial efficacy, poor selectivity for different Cxs, and significant effects on other cellular processes [165].

### 6.2. In Vivo Seizure Models

The strongest evidence linking Cxs and seizures is the anticonvulsant effect of agents that inhibit Cx channels in in vivo experimental models. In vivo antiseizure properties were assigned to CBX in chemical [95,102,113,166,167,168,169,170], electrical [171], and genetic animal models [146,154,172,173] (Table 5).

Since CBX has mineralocorticoid properties as well, several studies used control experiments with glycyrrhizin [172,173,174] or spironolactone (mineralocorticosteroid antagonist; co-administered with CBX) [151,166,175] to check whether the observed effect of CBX was due to its Cx channel blocking effect.

Among the fenamates, both meclofenamic acid and flufenamic acid have been tested in in vivo seizure models. Disease-modifying effects of meclofenamic acid were reported as it delayed kindling acquisition in both the amygdala and the rapid hippocampal kindling model in mice [97]. It also slowed down the onset of pentylenetetrazol (PTZ)-induced convulsions [176], but at the same time, potentiated its excitatory effects. In rats with an epileptic focus induced by tetanus toxin, a significant reduction in percentage of seizure time was observed with locally applied meclofenamic acid [169]. In the maximal electroshock (MES) model, meclofenamic acid (100 mg/kg i.p.) protected from tonic hind limb extension. A higher dose (200 mg/kg i.p.) was proconvulsive and toxic [162]. However, diclofenac, a structurally related compound of meclofenamic acid and well-known anti-inflammatory drug, also exhibited anticonvulsant activity in the same model [162]. The non-selective GJ enhancer trimethylamine is proconvulsive in distinct rodent models [95,98,154,170,171] (Table 5).

Overall, non-specific Cx channel blockers improve seizure outcome, while GJ enhancers exacerbate seizure outcome. Again, given the non-specificity of these agents and associated non-channel effects [52], conclusions from these studies should be taken with caution. Systemically administered, they can affect vital peripheral organs such as the heart [177,178].

## 7. Effects of Cx Mimetic Peptides in Rodent Models of Seizures and Epilepsy

### 7.1. Cx Mimetic Peptides

Most recent investigations aimed at specific targeting of astrocytic Cx43 channels as an approach towards AED discovery [179]. For this purpose, Cx43 mimetic peptides, i.e., short peptides identical to specific sequences of the Cx protein were used as tools to inhibit Cx43 channels in a more specific way. Cx43 mimetic peptides identical to sequences on the EL have been used to interfere with GJ formation during the docking of two HCs, a process that relies on well conserved EL sequences [49,180]. Two examples of such EL-mimetic peptides are Gap26 (VCYDKSFPISHVR) and Gap27 (SRPTEKTIFII). These peptides interact with yet unknown EL-located sequences and decrease GJ coupling [181]; Gap27 also affects Cx43 phosphorylation [182]. Later, it was found that Gap26/27 also inhibit Cx43 HCs [183,184] and it is now clear that Gap26/27 more rapidly inhibit HCs than GJs (a further discussion on these peptides can be found in [185,186]. Another mimetic peptide of EL sequences is Peptide5, which is shifted five amino acids in N-terminal direction compared to Gap27 (VDCFLSRPTEKT). Peptide5 inhibits HCs at low concentration (~5 µM), while higher concentrations (~500 µM) block GJs and HCs [187]. The above-mentioned Cx mimetic peptides are promising GJ/HC blocking tools, although there are some obstacles to their use. First, Gap26/27 are not specific for Cx43 and also inhibit vascular Cx37 and Cx40-based channels, which relates to the well conserved nature of the EL-located docking sequences (discussed in [185,186]). Second, at current, the effect of Gap26/27 peptides has not been tested for astrocytic Cx30 channels. Third, Cx mimetic peptides may also block Panx channels via physical steric hindrance when used at high concentrations [188,189]. On the other hand, intracellularly acting Cx mimetic peptides have intrinsically less chance of interacting with other Cxs, because the intracellular domains are the least conserved domains between distinct Cx isotypes, yet it cannot be excluded. Wang et al. were the first to report on selective inhibition of Cx43 HC by Gap19 (KQIEKKFK), a nonapeptide derived from the L2 domain located in the Cx43 CL (amino acids 128–136). The L2 domain and Gap19 sequence within this domain are important for CL interactions with the CT tail of Cx43 [190], involving the last nine amino acids of the CT (CT9) and the Src homology 3 (SH3) domain of the CT [48,191]). CT-CL interactions oppositely affect Cx43 GJs and HCs and act to close GJs while permitting HC to open in response to electrical or chemical stimulation [48,192,193]. Gap19 has no effect on Cx40 HC or Panx1 channels [194] but several other CNS Cxs still have to be tested before concluding Cx43 selectivity. In general, to improve membrane (and BBB) permeability for in vivo use, peptides can be coupled to cell-penetrating peptides, e.g., transactivator of transcription (TAT)-sequence of the human immunodeficiency virus-1 (HIV-1), or Xentry, derived from the X-protein of hepatitis B virus [195,196,197,198]. However, TAT raises safety concerns due to its high positive charge and low cell specificity causing cytotoxicity in in vivo studies [198]. Interpretation of research results can be complicated too, as TAT increases Cx43 mRNA and protein expression as well as GJIC in human astrocytes, by binding of TAT to the Cx43 promotor (chronic TAT incubation 6–48 h on cultures) [199]. Effects were specific to human cells, since treatment of mouse astrocytes with HIV-TAT did not alter Cx43 expression or GJIC [199]. As far as we know, it has not yet been demonstrated whether TAT is cleaved off from the coupled mimetic peptide in vivo. To rule out non-specific effects of cell-penetrating peptides like TAT or others, these sequences can be used without fusion to the Gap19 moiety as a control peptide to rule out non-specific effects [10,200]. TAT peptide was reported to increase several inflammatory markers in non-brain endothelial cells, including IL-1β, IL-6, monocyte chemoattractant protein 1 (MCP-1), and c-reactive protein (CRP) [200]. Further investigations can give answer to the question whether TAT-linked peptides can be used in clinic or whether other cell-penetrating peptide linkers might be more appropriate. A recent study suggests that Xentry is a safe cell-penetrating peptide facilitating the uptake of Gap19, at least in an in vitro setting [198]. 

Another Cx43 mimetic peptide, TAT-Cx43_266–283_, corresponds to a part of the sequence of the SH3 domain of the CT of Cx43, and has been shown to inhibit non-receptor tyrosine kinase c-Src activity, which plays an important role in several signaling pathways related to neuroinflammation [12]. Iyyathurai et al. recently identified the SH3 binding domain, together with the previously identified CT9 region, as a critical determinant for CT-CL interactions in Cx43 HCs, whereby TAT-SH3 (amino acids 273–285) synthetic peptide removes the high [Ca^2+^]_i_ brake on HC opening, thus acting as a Cx43 HC enhancer [191]. Thus, the Cx43_266–283_ mimetic peptide seems to inhibit Cx43 HCs while the Cx43_273–285_ peptide that overlaps with the previous peptide by 10 amino acids enhances HC activity. Possibly, the two peptides act on different aspects of Cx43, with TAT-SH3 (Cx43_273–285_) acting at the channel level and TAT-Cx43_266–283_ acting at the level of Cx43 turnover. Critical overviews on the use of Cx-targeting peptides is provided in [16,185,186,201]; and in particular, targeting Cx43 in [16].

### 7.2. In Vitro Seizure Models 

Braet et al. first demonstrated that Gap26 (30 min incubation) inhibits intercellular Ca^2+^ waves induced by inositol trisphosphate (IP_3_) in confluent endothelial cell cultures, however, not as a result of blocking GJIC as measured by FRAP, but due to selective inhibition of IP_3_-triggered ATP release, suggesting a role for Cx HCs [183]. Likewise, Torres et al. confirmed that Cx43 HCs participate in astrocytic Ca^2+^ wave propagation by releasing ATP in the extracellular space [56]. In line with this, Gap26 decreases basal extracellular Cx43-HC mediated ATP release in hippocampal slices, which influences hippocampal excitatory transmission [40]. Acute inhibition of astrocytic Cx43 HCs with Gap26 resulted in a rapid decrease in excitatory postsynaptic currents (EPSCs) from hippocampal CA1 pyramidal cells. Glial hypersynchronization and cell-–cell propagating glial Ca^2+^ waves are linked to epilepsy and involve Cx43 GJs as well as HCs. Interestingly, HC inhibition is sufficient to block intercellular Ca^2+^ waves [48,191] allowing to target Cx43 HCs without affecting GJs that would otherwise interfere with crucial physiological functions of these junctional channels. As a result, Cx43 HCs may represent a more appropriate AED target than Cx43 GJs, taking into account the ambiguous role of the latter in seizures. In experimental epilepsy research, Gap27 was the first Cx43 mimetic peptide that was tested in vitro for its effects on epileptiform activity. In organotypic hippocampal slice cultures from post-natal day 7 (PD7) rats, Gap27 inhibited spontaneous recurrent epileptiform activity, but only after prolonged (>10 h) treatment [14], suggesting the peptide’s action may also involve diminished Cx43 GJIC. Yoon and colleagues used peptide5 and found that low concentrations (5–50 µM; sufficient to block Cx43 HCs only) prevented neuronal damage in hippocampal slices during a 48 h treatment with bicuculline (GABA_A_ receptor antagonist). Oppositely, higher doses of peptide5 (500 µM; inhibiting HCs and GJs composed of Cx43 and possibly other Cxs) exacerbated the seizure-induced lesions. By contrast, all doses protected the CA1 region from further damage when peptide5 was applied during the recovery period. The authors therefore concluded that GJIC is essential for tissue survival when slices undergo stress due to enhanced excitability, while excessive HC opening might be deleterious. During the recovery period on the other hand, GJIC plays a crucial role in the spread of neuronal damage [13]. The two latter studies highlight again the dual role of Cx43 GJs in seizure-like events, while the study by Yoon et al. suggests detrimental effects of Cx43 HC opening [13].

Bazzigaluppi and coworkers examined the role of Cx43 GJs and HCs in [K^+^]_e_ regulation and seizure generation in the mouse neocortex in vivo, by using Gap27 and TAT-Gap19 [196]. When Gap27 was microinjected in the cortex, [K^+^]_e_ levels markedly increased from 1.9 ± 0.7 to 9 ± 1 mM, although seizures were not induced. By contrast, TAT-Gap19 did not affect [K^+^]_e_, suggesting that Cx43 HCs do not play a role in spatial K^+^ buffering; the results with Gap27, which rapidly inhibits HCs and more slowly GJs, suggest that GJ inhibition indeed interfered with spatial K^+^ buffering. When 4-aminopyridine was microinjected, [K^+^]_e_ increased to ~10 mM and spontaneous seizures were induced. Direct injection of 50 mM K^+^ solution into the parenchyma on the other hand did not trigger seizure activity, suggesting that [K^+^]_e_ elevation alone is not sufficient for seizure induction. Nevertheless, one must keep in mind that Gap26/27 peptides may inhibit other Cxs in neurons or vascular cells. Taken together, these in vitro data indicate that TAT-Gap19 is an interesting tool to decipher the role of Cx43 HCs in seizure activity without interfering with spatial K^+^ buffering.

### 7.3. In Vivo Seizure Models

Our group recently found that TAT-Gap19 pretreatment exerts anticonvulsant effects in in vivo chemically- and electrically-induced rodent seizure models [10]. We showed that the chemoconvulsant pilocarpine opens astroglial Cx43 HCs in acute hippocampal slices of mice, which was inhibited by TAT-Gap19. In the focal pilocarpine in vivo rat and mouse model, TAT-Gap19 decreased seizure duration as well as D-serine levels in microdialysate samples. The anticonvulsant effect of TAT-Gap19 was abolished when it was co-administered with D-serine, suggesting that Cx43 HC inhibition protects against seizures by lowering extracellular D-serine levels. However, the involvement of other gliotransmitters cannot be excluded. Cx43 HC inhibition also decreased seizure duration and seizure severity score in an acute and chronic electrical seizure mouse model of refractory seizures. All together, these results indicate that Cx43 HCs seem associated with ictogenesis since pretreatment with TAT-Gap19 acts in an anticonvulsive manner. Interestingly, we were unable to rescue full-blown seizures with TAT-Gap19 in the focal pilocarpine mouse model, suggesting that pretreatment with TAT-Gap19 is effective but treatment during ongoing seizures is not. Cx43 HCs may thus be involved in ictogenesis, while full blown seizure activity is likely to involve a large array of factors. In fact, this is a common feature for the majority of AEDs on the market, since most of them are prophylactic and aim to prevent seizures from happening [10].

Previous studies have shown that excitotoxic cortical lesions induced by kainic acid in vivo in adult mice, induce an increase of glial fibrillary acidic protein (GFAP) and a decrease of Cx43 expression in astrocytes [202], which correlates with a transient increase in c-Scr activity in the region surrounding neuronal lesion [11]. Gangoso et al. used TAT-Cx43_266–283_, a Cx43 mimetic peptide that inhibits c-Src activity, and found it to diminish neuronal cell death and reactive gliosis after cortical kainic acid injection in vivo in adult mice, compared to non-treated or TAT-treated animals [12]. The proposed mechanism of action of TAT-Cx43_266–283_ was further explored in vitro and was shown to inhibit astrocytic Cx43 HCs [12].

## 8. Conclusions

Currently, there is no AED on the market with Cx channel blockade as main mechanism of action. However, a marketed AED that modulates synaptic vesicle protein SV2A, levetiracetam, can counteract inflammation-induced Cx43 GJIC impairment [78] and astrocytic uncoupling [85]. The voltage-dependent Na^+^ channel-inhibiting AEDs, carbamazepine, lacosamide, and zonisamide, can modulate astroglial Cx43 HC function and gliotransmission [75,76].

We think that Cx43 HCs are an interesting and promising antiepileptic drug target since selective Cx43 HC inhibition seems to lead to promising anticonvulsive effects, while Cx43 GJ inhibition can have both pro- and anticonvulsive actions. In line with this, the role of Cx43 GJs in other pathologies has been summarized as “controversial” [203,204]. For instance, in stroke, GJIC may provide neuroprotection by allowing antioxidants, ATP, glucose, and others to move into areas of high energy demand, while also diluting cytotoxic levels of excitatory amino acids and ions through the astrocytic syncytium (“good Samaritan effect”) [203,205,206,207,208]. On the other hand, GJIC may extend cellular injury through intercellular propagation of cytotoxic substances [209]. The same might hold true for Cx43 GJ inhibition in epilepsy. Moreover, since Cx43 GJs are ubiquitously present in the human body and often seen as “physiological pores” and Cx43 HCs as “pathological pores”, it is reasonable to believe that Cx43 HC inhibition will be endowed with less potential side-effects compared to GJ inhibition. Indeed, Cx43 is the main Cx in the heart, in particular, in ventricular cardiomyocytes. Myocardial Cx43 GJs facilitate (physiological) electrical cell–cell coupling that is essential for impulse propagation, while uncontrolled Cx43 HC opening may be deleterious for the cells as it may pass electrical leakage current and provoke ionic imbalance, cell swelling, and loss of essential metabolites [194]. As such, in the heart, Cx43 GJIC blockade may have potentially dangerous pro-arrhythmogenic side effects by disturbing electrical coupling. Therefore, selective Cx43 HC inhibition for antiseizure purposes may be a safer option to prevent such deleterious effects. Hemichannel blocking peptides that interact with intracellular targets of the Cx43 protein in this sense offer the advantage of increased Cx isotype specificity, given the less conserved nature of intracellular sequences. However, a thorough screening of effects of Cx mimetic peptides on all Cx isotypes (and other targets) is necessary before claiming selectivity of a given peptide. Furthermore, the Cx43 HC inhibitor should maintain cell–cell communication. Noteworthy, this is facilitated by the fact that Cx43 HCs and GJs are often inversely modulated. Nevertheless, Cx43 HCs may also be involved in physiological processes [210] and their inhibition might cause memory impairments, as recently described in (non-epileptic) mice [47,211]. Unfortunately, memory impairment is among the most common complaints of patients with epilepsy and multiple AEDs on the market significantly reduce memory performance [212]. Further studies will be needed to obtain a clear view on the safety profile of selectively targeting Cx43 HCs.

One major drawback of peptide therapeutics is their short plasma half-life, negligible oral bioavailability [213], low permeability and solubility and poor absorption-distribution-metabolism-excretion (ADME) properties [214]. The synthesis of stable, systemically active and CNS penetrant analogues should therefore be continued. Currently, peptides only represent 2% of the drug market [214], and approximately 75% of peptide drugs are administered parenterally [215], which is a less appealing option for patients, especially for chronic therapies such as for epilepsy. Intranasal administration, where drugs are directly delivered into the brain via the olfactory route, might overcome low brain permeability problems, reduce drug degradation and wastage of drug through systemic clearance [216]. Nevertheless, it remains to be investigated whether Cx43 mimetic peptides as such can be used in clinical settings. 

Noteworthy, Cx43 emerges as a possible target in multiple non-CNS diseases (e.g., vascular diseases, wound healing diseases, retinopathies, cardiac diseases, cancer, and chronic kidney, liver, and lung diseases) and brain diseases (Alzheimer’s disease, epilepsy, and stroke) and several agents that inhibit Cx43 channels are currently investigated as potential clinical therapeutics by various pharma and biotech companies [217]. FirstString Research, USA, focusses on Cx43 as a therapeutic target in scar prevention, inflammation reduction, wound healing, and tissue regeneration and implemented a 25 amino acid peptide αCT1. This CT-targeting peptide induces transition of cell-surface Cx43 from HCs to GJs, enhances the stability of GJ aggregates, reduces the plasma membrane HC pool that is prone to activation, and tempers inflammatory responses [218]. A topical αCT1 formulation (Granexin^®^) has entered clinical phase II and III trials (against diabetes foot ulcers, venous leg ulcers, surgical incision scars, radiation injuries) [219,220,221]. Zealand Pharma, Denmark, possesses several libraries of GJ modulating compounds, including a library of Cx43 CT interacting compounds and peptides (e.g., danegaptide, a GJ enhancer with oral bioavailability [222]); and focus on accelerated dermal wound healing. Noteworthy, danegaptide increases astrocytic Cx43 coupling with no significant effects on Cx43 HC activity in vitro and decreases infarct volume in a mouse brain ischemia/reperfusion model [223]. Interestingly, CoDa Therapeutics Inc., USA and New Zealand, has Cx channel modulators in clinical development (chronic skin wounds, ocular disease), e.g., an antisense oligodeoxynucleotide (Nexagon^®^) that transiently downregulates Cx43 protein expression (e.g., thermoreversible gel for eye injuries [224]), an extracellular acting Cx43 peptidomimetic that can be delivered locally or systemically (Peptagon™ [225]) (Peptide5) and a small molecule for systemic or oral delivery (HCB1019). The latter two are specifically used to target HCs; Nexagon^®^ and HCB1019 are Phase III ready. We hypothesize that these compounds might be able to reduce inflammation following traumatic brain injury as well, which is one of the common causes of epilepsy. Overall, the above overview demonstrates growing interest in Cx43 channel modulators for use in a wide variety of tissue and organ targets; as such we believe that exciting times are ahead for Cx43 HC modulation in the field of epilepsy.

## Figures and Tables

**Figure 1 biomolecules-10-01578-f001:**
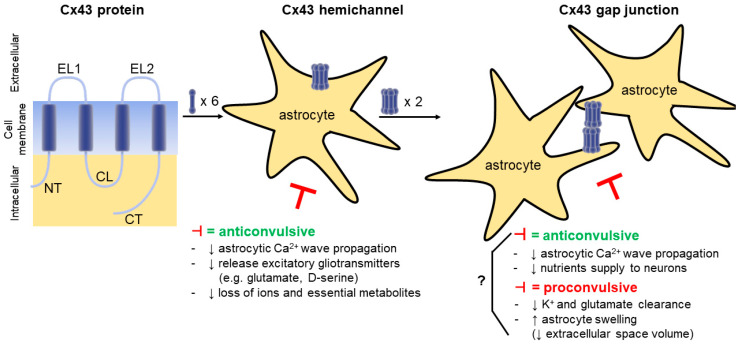
Schematic diagram illustrating connexin43 (Cx43) protein, hemichannel (HC), and gap junction (GJ) and their involvement in seizures. (**Left**) Each Cx43 protein has four transmembrane domains, two extracellular loops (EL1-2), one cytoplasmic loop (CL), and C- and N-terminal tails (CT and NT). Its mRNA, protein expression levels, phosphorylation state, and distribution can be altered in experimental seizure models and tissue from patients with epilepsy. (**Middle**) A Cx43 HC is built up by six Cx43 proteins and forms a channel between the astrocytic cytosol and the extracellular environment. Cx43 HC inhibition (⊣) can be anticonvulsant by decreasing (↓) astrocytic Ca^2+^ wave propagation, release of excitatory gliotransmitters and loss of ions, essential metabolites. (**Right**) A Cx43 GJ is formed by docking of two Cx43 HCs from adjacent astrocytes. Cx43 GJ inhibition (⊣) can be both pro- and anticonvulsant. Ictogenic properties originate from decreased (↓) K^+^ and glutamate redistribution and increased (↑) astrocyte swelling; anticonvulsant properties are due to decreased (↓) astrocytic Ca^2+^ wave propagation (hypersynchronization) and trafficking of nutrients to neurons.

**Table 1 biomolecules-10-01578-t001:** Cx43 expression and phosphorylation changes associated with human epilepsies.

Epileptic Condition	Brain Region (vs. Control Tissue)	∆ mRNA Expression	∆ Protein Expression, Phosphorylation
Intractable seizures of structural etiology [65] Cerebral tumors with acute seizures [65]	Temporal lobe neocortex Neocortex (vs. cerebral tumors without seizures)	↑ ↑	
Complex partial seizures (mesial temporal lobe epilepsy (MTLE)) (structural etiology) [73]	Hippocampus (vs. temporal lobectomy)	↓	=
Epilepsy associated brain tumors [66]	Brain tumor and perilesional epileptic cortex (vs. normal cortex)		↑ in low-grade gliomas, perilesional cortex (≠ isoforms); ↓ in high-grade gliomas (P0 isoform)
Generalized seizures in progression of MTLE (structural etiology) [70]	Hippocampus (vs. post-mortem tissue)		↑ CA1 and CA4
MTLE (structural etiology) [68]	Hippocampus (vs. post-mortem tissue)		↑
TLE (structural etiology) [65,67]	Hippocampus (vs. post-mortem tissue)		↑
Cryptogenic epilepsy or epilepsy secondary to focal cortical dysplasia (FCD) (structural etiology) [71]	Cortex (vs. tissue resected during tumor surgery and autopsy tissue)	↑ in 25% of cryptogenic epilepsy	↑ in FCD type IIB (large Cx43 aggregates around balloon cells and astrocytes); = in cryptogenic epilepsy and FCD type IA/IIA
Refractory epilepsy of structural etiology [72]	Epileptic foci (vs. tissue traumatic brain injury)		↑
MTLE-hippocampal sclerosis (HS) (structural etiology) [74]	Sclerotic hippocampus (MTLE-HS vs. MTLE non-HS tissue)		↑ total (whole cell); = in plasma membrane, large Cx43 plaques around blood vessels (↑ number and size), ↑ phosphorylation S255

∆, changed; =, unaltered; ≠, different; ↑, increase(d); ↓, decrease(d); CA, cornu ammonis; FCD, focal cortical dysplasia; HS, hippocampal sclerosis; (M)TLE, (mesial) temporal lobe epilepsy.

**Table 2 biomolecules-10-01578-t002:** Cx43 expression and phosphorylation changes in the hippocampus in rodent models of seizures and epilepsy.

Seizure/Epilepsy Model	∆ mRNA Expression			∆ Protein Expression, Phosphorylation, Coupling		
**Ex vivo epileptiform Activity**						
Repetitive tetanization of Schaffer collaterals in CA1 (slices) from post-natal day (PD)25–40 rats [110]				= total, ↓ non-phosphorylated (P0) (redistribution from P0 to phosphorylated (P))		
Co^2+^ in whole hippocampal isolates from PD15 mice [62]	↑			P1 ↑ P2 and P0 slightly ↑		
*C*hronic (18 h) bicuculline in organotypic hippocampal slice cultures of PD7 rats [96]	↑			↑ in membrane fractions also ↑ coupling		
**In Vivo Models**						
	**Acute**	**Latent**	**Chronic**	**Acute**	**Latent**	**Chronic**
**Status epilepticus (SE) Models**						
I.p. Li^+^-pilocarpine SE in rats [98]				↑ 2–12 h, most at 24 h post-SE (CA1-3)		= 30 d and 60d post-SE
I.p. pilocarpine SE in rats (200–250 g) [95]				↑ after focal seizures		
I.p. pilocarpine SE in rats (270–300 g) [114]				total =, but ≠ distribution 2 h post-SE: ↑ str.pyramidale, ↓ str.radiatum	total ↑, but = distribution 3 d post-SE	
I.p. lithium-pilocarpine SE in rats (180–200 g, 6 w old) [108]				Total ↓ 24 h post-SE, gradually ↑ from 7 to 60 d after SE; ↓ P-Cx43 (S368) 24 h post-SE, gradually ↑ from 7 to 30 d after SE, peak at 60 d		
I.p. pilocarpine SE in mice (25–30 g) [100]	= 4 h, 1 d post-SE	↑ 1 w post-SE	↑ 2 m post-SE	= 4 h and 1 d post-SE	↑ 1 w post-SE	↑ 2 m post-SE
I.p. pilocarpine SE in mice (22–28 g, 8 w old) [102]				↑ 3 h, peak between 1 and 3 d post-SE	↑ 7 d, ↑ 15 d post-SE	= as baseline 30 d post-SE
I.p. kainic acid (KA) SE in rats (150–180 g) [99]					↑ 1 w post-SE, also ↑ coupling	
I.p. KA SE in PD26-33 rats [112]			= 4 w post-SE			= 4 w post-SE
Intracortical KA in mice (3–4 m old) [72]						3 m post-SE: ↑ total, ↑ P2 (whole-cell); = plasma membrane, shift to P2, Cx43 accumulates around blood vessels
I.c.v. KA in rats (280 ± 300 g) [103]	24 h: ↓ CA3-4 pyr.layer, ↑ other regions	48–72 h: ↓ CA3-4 pyr.layer, ↑ other regions			48 h: ↑ layer oriens, molecular, lacunosum molecular	
**Chemical kindling models**						
I.p. pentylenetetrazol (PTZ)-kindled rats (180–200 g) [91]						↑ CA3 (after 2 w, in fully kindled rats)
**Electrical kindling models**						
Amygdala kindling in PD83-95 rats (230–250 g) [112]			slight↓/= 2–6 w after last stage 5 seizure			= 2–6 w after last stage 5 seizure
Amygdala kindling in rats (300–350 g) [109]		= 24 h after stage 2 seizures	= 24 h after stage 5 seizures		= 24 h after stage 2 seizures	= 24 h after stage 5 seizures
Hippocampal kindling in adult rats (275–300 g) (escalating stimulations until after discharges) [111]	= after 3 h, 12 h, 24 h			= after 3 h, 12 h, 24 h		
**Other seizure/epilepsy models**						
PTZ in rats (14 w old) [101]				↑ after 2 h, most 8 h		
Genetic mouse model of tuberous sclerosis complex [Tsc1^GFAP^CKO mice (gene Tsc1 inactivated in glia)] [107]						↓ in mice 2–3 w old (precedes seizure onset) and mice 4–5 w (seizure onset) (also ↓ coupling)
Experimental febrile seizures induced by hyperthermia (HT) in PD14-15 mice [115]					5 d post-HT: total ↓, ↑ P1/P2 (↓coupling)	
Chronic i.c.v. administered lipopolysaccharide (LPS) in adult rats (280–320 g) [106]	=	=	=	=	↓ after 7 LPS injections	=
Intrahippocampal 4-aminopyridine (4-AP) in rats (250–300 g) [94]	trend towards ↑ after 1 h			↑		

∆, changed; =, unaltered; ≠, altered; ↑, increase(d); ↓, decrease(d); 4-AP, 4-aminopyride; CA, cornu ammonis; d, days; i.c.v., intracerebroventricular; i.p., intraperitoneal, h, hours; HT, hyperthermia; KA, kainic acid; LPS, lipopolysaccharide; m, months; P(1/2), phosphorylated isoform (1/2); PD, post-natal day; PTZ, pentylenetetrazole; SE, status epilepticus; w, weeks.

**Table 3 biomolecules-10-01578-t003:** Cx43 expression and phosphorylation changes in other seizure-relevant regions in rodent models of seizures and epilepsy.

Seizure/Epilepsy Model	∆ mRNA Expression	∆ Protein Expression, Phosphorylation
Amygdala kindling in rats (180–250 g) [109]	↓ in amygdala after 4 w, but normalization with increasing numbers of stimulations	same as mRNA
Tetanus-toxin in amygdala of rats (180–250 g) [109]	↓ or = in amygdala after 4, 8, 10 w	same as mRNA
Tetanus-toxin in amygdala of rats (250–320 g) [110]	↓ or = in amygdala and posterior cerebral cortex (no distinct temporal profile of Cx43 expression during epileptogenesis)	
Local in vivo 4-aminopyridine (4-AP) in post-natal day (PD)40–50 rats [102]	↑ in somatosensory cortex (primary focus (Pf) and mirror focus (Mf)) (60 min after intensive seizure activity)	
Local in vivo 4-AP in PD30-40 rats [95]	↑ in Pf and Mf (60 min after seizure onset)	
I.p. 4-AP in adult rats (200–250 g) [118]	= 1, 3, 24 h in neocortex after seizure induction	same as mRNA, 50% ↓ in phosphorylated (P1, P2) to non-phosphorylated (P0) Cx43 at 3 h
FeCl_3_ injection in sensimotor cortex in frontal lobe of rats (6–8 w) (model posttraumatic epilepsy) [119]	↑ in cortex after 14 d	↑ in cortex after 14 d
Pentylenetetrazol in rats (14 w) [106]		↑ in cortex after 8 h
Genetic model of tuberous sclerosis complex [Tsc1^GFAP^CKO mice of 4–5 w] [112]		↓ in neocortex
Genetic rat model of autosomal dominant sleep-related hypermotor epilepsy (ADSHE) with S284-L mutation [76]		↑ in thalamus and frontal cortex

∆, changed; =, unaltered; ↑, increase(d); ↓, decrease(d); 4-AP, 4-aminopyride; ADSHE, autosomal dominant sleep-related hypermotor epilepsy; i.p., intraperitoneal, h, hours; Mf, mirror focus; P0, non-phosphorylated; P(1/2), phosphorylated isoform (1/2); PD, post-natal day; Pf, primary focus; w, week.

**Table 4 biomolecules-10-01578-t004:** Effects of non-selective gap junction blockers in in vitro seizure models.

Experimental Model	Slices/Animals	Blocker(s)	Results
Repetitive tetanization of Schaffer collaterals [115]	Entorhinal cortex-hippocampal slices from post-natal day (PD)25–40 rats	octanol, CBX, sodium propionate	↓ duration of seizure-like primary after discharges (PADs) in CA1 pyramidal region
Single stimulus and brief tetanic Schaffer collateral stimulation [14]	Organotypic hippocampal slice cultures of PD7 rats, cultures used after 10–14 d	CBX	↓ spontaneous and evoked seizure-like events (SLEs)
Mg^2+^-free induction of epileptiform activity [148]	Hippocampal slices of rats (90–350 g)	halothane, octanol, CBX	intact primary bursts (i.e., typical interictal burst), ↓ secondary discharges (i.e., lasting for several 100 ms)
Mg^2+^-free induction of epileptiform activity [155]	Neocortical slices of rats (5–8w)	CBX	↑ frequency and amplitude of SLEs
Mg^2+^-free induction of epileptiform activity [147]	Entorhinal cortex-hippocampal slices from PD12–14 rats	CBX Cx43 antibody (Abcam, #ab11370)	7/17 slices: complete ⊣ of SLE 10/17 slices: ↑ interictal interval 5/12 slices: complete ⊣ of SLE 7/12 slices: ↑ interictal interval
Mg^2+^-free induction of epileptiform activity [154]		CBX Cx43 antibody (Abcam, #ab11370)	Inhibition or complete ⊣ of SLE (SLE only in 2/12 slices but ↓ duration) 5/5 slices: complete ⊣ of SLE
Ca^2+^-free induction of field burst activity [156]	Hippocampal slices of PD20-30 rats	sodium propionate, octanol, halothane	↓ epileptiform activity in CA1 area
High K^+^/low Ca^2+^-induction of evoked and spontaneous epileptiform field potentials [150]	Hippocampal slices (CA3 area) of rats (+/−200 g)	octanol, heptanol CBX	↓ spontaneous field bursting, ⊣ all epileptiform markers of the evoked responses dose-dependent anti-bursting activity, = number of repetitive population spikes evoked by single stimuli
High K^+^/low Ca^2+^-induction of evoked and spontaneous epileptiform field potentials [152]	Hippocampal slices (dentate gyrus area) of adult rats (100–350 g)	octanol, oleamide and CBX	⊣ prolonged field bursts
4-AP induced spontaneous ictal-like activity (ILA) [146]	Thalamocortical slices of adult genetic absence epilepsy rats from Strasbourg (GAERS) or non-epileptic rats (NER)	CBX	↓ frequency and ↓ duration of ILA, but less rapidly in GAERS than in NER slices
Bicuculline (18 h)-induced epileptiform activity [101]	Organotypic hippocampal slices (CA1 str. pyramidale) of PD7 rats	CBX	⊣ spontaneous and evoked epileptiform discharges reversibly
Bicuculline (2–10 h)-induced epileptiform activity [149]	Whole hippocampus of PD15 mice	CBX	⊣ spontaneous epileptiform discharges
Bicuculline-induced epileptiform activity [157]	Piriform cortex; isolate brain preparation of young adult guinea pigs (200–250 g)	octanol, 18-alpha-glycyrrhetinic acid	⊣ spontaneous interictal spikes
Co^2+^-induced epileptiform discharges [64]	Whole hippocampus isolates from PD15 mice	Octanol	⊣ ictal-like and interictal discharges

↑, increase(d); ↓, decrease(d); ⊣, block; 4-AP, 4-aminopyridine; CA, cornu ammonis, CBX, carbenoxolone; d, days; GAERS, genetic absence epileptic rats from Strasbourg; ILA, ictal-like activity; NER, non-epileptic rats; PADs, primary after discharges; PD, post-natal day; SLEs, seizure-like events; w, weeks.

**Table 5 biomolecules-10-01578-t005:** Effect of non-selective gap junction blockers and openers in in vivo rodent models of seizures and epilepsy.

Experimental Model	Animals	Read-Out	Blocker/Opener(s)	Results
**Chemically-Induced Seizure Models**				
4-aminopyridine (4-AP) in entorhinal cortex and hippocampus [168]	Adult rats (250–350 g)	Electroencephalogram (EEG) + behavior	CBX (in entorhinal cortex, 30 min after 4-AP)	↓ amplitude and frequency of epileptic discharges epileptic trains: ↓ number and duration both in injected entorhinal cortex and in propagated CA1, no convulsive behavior
4-AP in entorhinal cortex and CA1 hippocampus [98]	Adult rats (250–350 g)	EEG + behavior	trimethylamine (locally)	no proconvulsive synergistic effect, however: trimethylamine + 4-AP produced seizure activity patterns with continuous, long epileptic discharges of ↑ amplitude and ↓ frequency (during first 30 min)
4-AP in somatosensory cortex [95]	Anaesthetized post-natal day (PD)30–40 rats (200–250 g)	Electrocorticography (ECoG) (in primary focus (Pf) and mirror focus (Mf))	CBX (at already active Pf, after 4-AP) trimethylamine (at already active Pf, after 4-AP)	↓ seizure duration, ↓ amplitude of seizure discharges (maximal after 10 min) ↑ seizure duration, ↑ frequency and amplitude of epileptic discharges
4-AP in neocortex [102]	Anaesthetized PD40–50 rats (200–250 g)	ECoG (in Pf and Mf)	CBX (after 4-AP) CBX (before 4-AP)	↓ seizure intensity of already active epileptic foci mild influence on ictogenesis
Pentylenetetrazol (PTZ) i.p. [167]	Adult mice (25–30 g)	seizure onset, duration, mortality	CBX (i.p.)	↑ onset time of seizures, ↓ seizure duration
Tetanus toxin in motor cortex (refractory cortical epilepsy model) [169]	Freely moving awake rats (240–320 g)	EEG + electromyogram (EMG)	CBX and meclofenamic acid (locally at seizure focus)	↓ % of seizure time (spontaneous minor motor seizures)
i.c.v. penicillin [166]	Awake rats (4 m; 190–250 g)	EEG + behavior	CBX (i.c.v.)	↓ amplitude and frequency of epileptic spikes, ↓ epileptic behavior
Model of atypical absence seizures [170]	Long-Evans rat pups given s.c. cholesterol synthesis inhibitor AY9944 every 6d from PD2–22; rats used at P55-90	local field potentials recordings	CBX (in nucleus reticularis thalami (NRT)) (in hippocampus) trimethylamine (in NRT)	↓ seizure duration ↓ seizure activity ↑ seizures and spindle activity
Electrically-induced seizure models				
Amygdala kindling [171]	Adult rats (300–350 g)	EEG (after discharge (AD) in basolateral amygdala (BLA))	CBX (in BLA) trimethylamine (in BLA)	↓ AD duration (ADD), ↓ generalized seizure stage 5 duration (S5D), ↓ rats with S5 ↑ seizure susceptibility, (↑ ADD and ↑ S5D)
Amygdala kindling [97] Rapid hippocampal kindling [97]	Adult mice (22–25 g)	EEG (AD threshold (ADT)) + behavior	meclofenamic acid (i.c.v. prior to daily stimulus)	↓ seizure stage, ↓ ADD, ↑ stimulations required to elicit stage 5 seizure, ↑ ADT ↓ kindling acquisition in hippocampus
Maximal electroshock (MES) model [162]	Adult mice	Protection (%) from tonic hind limb extension	meclofenamic acid (i.p. prior to electroshock)	50 mg/kg: 17% protection 100 mg/kg: 75% protection 200 mg/kg: proconvulsive, toxic
Genetic animal models				
Audiogenic seizures [174]	DBA/2 mice (PD22-26 and PD48-56)	behavior	CBX (i.p.)	↓ audiogenic seizure intensity, ↑ effect of anti-epileptic drugs when co-administered
Genetically epilepsy prone rats (GEPRs) [172]	Audiogenic epileptic strain from Sprague-Dawley rats (12–18 w)	behavior	CBX (i.v. or i.p.) Bilateral injection in inferior colliculi, substantia nigra and inferior olivary complex	↓ clonic and tonic phases, ↓ seizure severity score, ↑ latency onset time ↓ seizure severity score
Genetic model of absence epilepsy [173]	WAG/Rij rats	EEG	CBX (i.v., i.p.) CBX (bilateral injection in NRT and nucleus ventralis posterolateralis (VPL) thalami)	no effect ↓ duration and number of spike-wave discharges (SWDs)
Genetic model of absence epilepsy [154]	WAG/Rij rats	EEG	CBX (i.p.) trimethylamine (i.c.v)	↑ SWD number and total time, ↓ length of interictal interval ↓ SWD number, ↑ length of interictal interval
Genetic model of absence epilepsy [173]	Lethargic (*lh*/*lh*) mice	EEG	CBX (i.p., i.c.v.)	↓ number and duration of SWDs
Genetic model of absence epilepsy [146]	Genetic absence epilepsy rats from Strasbourg (GAERS)	EEG	CBX (i.p.)	↓ duration of cortical SWDs, unaltered SW amplitude/frequency

↑, increase(d); ↓, decrease(d); 4-AP, 4-aminopyridine; AD(D/T), after discharge (duration/threshold); BLA, basolateral amygdala; CA, cornu ammonis; CBX, carbenoxolone; ECoG, electrocorticogram; EEG, electroencephalogram; EM, electromyogram; GAERS, genetic absence epilepsy rats from Strasbourg; GEPRS, genetically epilepsy prone rats; i.c.v., intracerebroventricular(ly) i.p., intraperitoneal(ly); i.v., intravenous(ly); m, months MES, maximal electroshock; Mf, mirror focus; NRT, nucleus reticularis thalami; PD, post-natal day; Pf, primary focus; PTZ, pentylenetetrazol; s.c., subcutaneous(ly); S5(D), seizure stage 5 (duration); SWDs, spike-wave discharges; VPL, nucleus ventralis posterolateralis; w, week.

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
