# Peer review of "Astrocytic Connexin43 Channels as Candidate Targets in Epilepsy Treatment"

_biomolecules, 2020, doi:10.3390/biom10111578_

Round 1

Reviewer 1 Report

The manuscript entitled “Astrocytic connexin43 channels as candidate anti-seizure drug targets” by Walrave et al. summarizes an important and emerging issue, the involvement and therapeutic potential of Cx43 in epilepsy. The manuscript is well written, easy to follow and adequately and deeply summarizes all the important issues on the field. Considering the exhaustive details covering the full field of the subject, I feel that the manuscript can be published in its present form. I raise only two minor issues, what the authors could address.

First, although the focus of the topic is epilepsy, a bit more attention could be placed on off-brain targets, since the therapeutic potential of Cx43 inhibitors may be severely reduced due to cardiological side effects.

Second, the authors rightfully identify non-specificity of current HC and GJ inhibitors as the main objection to further development of the field. However, much less concern is expressed about the specificity of peptide mimetic inhibitors. At some points the authors imply that specificity of peptide mimetics significantly surpasses that of small molecule inhibitors (for example, page 15, line 455: “specific sequences of the Cx protein were used as tools to inhibit Cx43 channels in a more specific way.” In the same context, however, it is also acknowledged that peptide mimetics are based on conserved EL sequences, therefore their Cx subtype-specificity is not a priori given. To confirm any degree of Cx43 specificity (or either any degree of connexin-specificity!), experimental data on differential activity of mimetic peptides on various Cx subtypes (or other different targets) should be cited. Relying solely on a connexin-specific sequence, does not necessarily infer any level of specificity. Especially considering that these sequences are buried in the 3D structure of the connexin and therefore are not readily accessible for the peptide inhibitors.

Author Response

Dear reviewer,

We would like to thank you for the valuable and constructive comments regarding our manuscript. Please find here below our responses to your suggestions. We hope the adaptations meet your expectation and look forward to your decision.

Point 1            
First, although the focus of the topic is epilepsy, a bit more attention could be placed on off-brain targets, since the therapeutic potential of Cx43 inhibitors may be severely reduced due to cardiological side effects.

Response 1    
Thank you for this very relevant remark since indeed, despite being abundantly expressed on astrocytes in the brain, Cx43 is the main Cx in the heart, in particular in ventricular cardiomyocytes. Myocardial Cx43 GJs facilitate (physiological) electrical cell-cell coupling that is essential for impulse propagation, while uncontrolled Cx43 HC opening may be deleterious for the cells as it may pass electrical leakage current and provoke ionic imbalance, cell swelling and loss of essential metabolites (Wang et al. 2013, PMID 2318439). As such, in the heart, Cx43 GJIC blockade may have potentially dangerous pro-arrhythmogenic side effects by disturbing electrical coupling. Therefore, selective Cx43 HC inhibition for antiseizure purposes may be a safer option to prevent such deleterious effects. This was added to the conclusion in lines 613-619 of the revised manuscript.

Point 2
The authors rightfully identify non-specificity of current HC and GJ inhibitors as the main objection to further development of the field. However, much less concern is expressed about the specificity of peptide mimetic inhibitors. At some points the authors imply that specificity of peptide mimetics significantly surpasses that of small molecule inhibitors (for example, page 15, line 455: “specific sequences of the Cx protein were used as tools to inhibit Cx43 channels in a more specific way.” In the same context, however, it is also acknowledged that peptide mimetics are based on conserved EL sequences, therefore their Cx subtype-specificity is not a priori given. To confirm any degree of Cx43 specificity (or either any degree of connexin-specificity!), experimental data on differential activity of mimetic peptides on various Cx subtypes (or other different targets) should be cited. Relying solely on a connexin-specific sequence, does not necessarily infer any level of specificity. Especially considering that these sequences are buried in the 3D structure of the connexin and therefore are not readily accessible for the peptide inhibitors.

Response 2    
We agree with the reviewer that extracellularly acting Cx mimetic peptides are not Cx subtype specific as the EL domains are highly conserved between different Cx isotypes (as are the NT and TM sequences). On the other hand, sequence diversity between Cxs lies in the intracellular regions (the CL and CT region). We acknowledge that a thorough screening of effects of Cx mimetic peptides on other Cx isotypes is necessary before deciding the selectivity of a given peptide. Nevertheless, the specificity of Cx mimetic peptides depends on the region in the Cx sequence they mimic: 1/ Cx mimetic peptides that are based on the conserved regions (e.g. EL) are more “broad-spectrum” Cx mimetic peptides and target several Cxs, while 2/ Cx mimetic peptides targeting the least conserved intracellular domains (the CL, CT region) have a lower change of interacting with other Cxs than the one their sequence was based upon (Delvaeye et al. 2018, PMID 30424929). Gap19 is an example of an intracellularly targeting Cx mimetic (sequence based on the CL of Cx43) and this peptide has been tested on Cx40 and on Panx1 HCs where it was found to have no effects on HC function (Wang et al. 2013, PMID 23184389). In the revised manuscript, we added some extra wording to explain why intracellularly targeting peptides (e.g. Gap19) have better credentials in terms of targeting Cx43 compared to extracellularly targeting peptides (e.g. Gap26 and Gap27), cited the references with experimental data on differential activity of mimetic peptides on various Cx subtypes, and remarked that effects on other Cx isotypes cannot be excluded (lines 486-493; 499-500; 619-623).

Reviewer 2 Report

Major comment

This is a very comprehensive review of epileptic pathomechanisms and candidate potentials of pathophysiology associated to connexin43. In this review article, the authors have done to review published findings regarding the effects of hemichannel/gap-junction inhibitors on various epileptic and convulsive models in depth.

Providing that editors do not object to such an opinionated article, this reviewer wants to recommend it for publication, since this reviewer agrees with the main conclusion in this review article, and considers to be able to contribute to the greater scientific debate in this area.

Minor comments

1)The authors used "anti-seizure drug" (ASD), whereas I feel this word is not adequate with this article. If possible, I would like to request to author that they use the words properly among anticonvulsant, antiepileptic drug and anti-seizure drug.

2)The authors should use appropriate terminology about etiological epileptic class, i.e. structural epilepsy, genetic and/or inherited epilepsies.

3) The authors should also use appropriate terminology between epileptogenesis-ictogenesis, pathogenesis-pathophysiology, and human epilepsy-epileptic animal model-convulsive animal model.

4) In this manuscript, there are several discrepancies. Please describe more detail explanation, to avoid reader's confusion, i.e.  L66 vs L166 and L139 vs L167.

5) Please explain R0/P1/P2 Cx43 and their function (L257).

6) Please move some sentences regarding the "tonabersat" from conclusion section.

Author Response

Dear reviewer,

We would like to thank you for the valuable and constructive comments regarding our manuscript. Please find here below our responses to your suggestions. We hope the adaptations meet your expectation and look forward to your decision.

Point 1
The authors used "anti-seizure drug" (ASD), whereas I feel this word is not adequate with this article. If possible, I would like to request to author that they use the words properly among anticonvulsant, antiepileptic drug and anti-seizure drug.

Response 1    
We understand the remark of the reviewer but would like to explain why we chose the wording “antiseizure drug (ASD)” instead of “anti-epileptic drug (AED)”. It is true that medicines for epilepsy are often called AEDs, but these treatments only suppress seizures while having no known impact on the epilepsy itself. Therefore, it has been suggested to refer to pharmacological agents that have a merely symptomatic effect as ASDs (French et al. 2020, PMID 32077329). This change in (using appropriate) terminology is especially important since in epilepsy research, emphasis is currently put on exploring new compounds that might have disease-modifying and/or anti-epileptogenic effects (“true AEDs”). If the reviewer would agree, we would therefore opt to keep the wording “anti-seizure drug” to describe the marketed treatments for patients with epilepsy.

Point 2
The authors should use appropriate terminology about etiological epileptic class, i.e. structural epilepsy, genetic and/or inherited epilepsies.

Response 2    
We now named the epilepsies with their etiological epileptic classes (i.e. structural epilepsy, genetic and/or inherited epilepsies) in the revised manuscript, e.g. lines 136, 139-140, 156, 167, 191, 222, table 1, table 2.

Point 3
The authors should also use appropriate terminology between epileptogenesis-ictogenesis, pathogenesis-pathophysiology, and human epilepsy-epileptic animal model-convulsive animal model.

Response 3    
We changed terminology in the revised manuscript, e.g. lines 85, 126, 128-129, 185, 344, 366, 402, 438, 544, 548, 554, 556, 582, 585, table 4, table 5.

Point 4
In this manuscript, there are several discrepancies. Please describe more detail explanation, to avoid reader's confusion, i.e.  L66 vs L166 and L139 vs L167.

Response 4

  1. L66 vs. L166

L66 - Cx43 HCs have a single channel conductance of ~220 pS and open at low extracellular Ca2+ concentration ([Ca2+]e) or moderately increased intracellular Ca2+ concentration ([Ca2+]i) [27-30], extracellular alkaline pH [31], or by cytokines (e.g. tumor necrosis factor α (TNF-α) and interleukin 1-beta (IL-1β)) [32], oxidative stress [33], dephosphorylation in ischemia [34], mechanical stimulation [35-37] and others.             

L166 - Levetiracetam reversed the downregulation of Cx43 expression and impaired GJIC in astrocytes co-cultured with 30% microglia (i.e. mimicking inflammation) [76].

We agree with the reviewer that L66 and L166 are seemingly discrepant (Cx43 HCs opened by cytokines (inflammation) vs. Cx43 GJIC impaired due to LPS (inflammation)) but this can be explained by the opposite regulation of Cx43 HCs vs Cx43 GJs. Following phosphorylation, GJs can undergo conformational changes in order to modify channel gating as described in the “ball-and-chain” model. Phosphorylation enables the CT to interact with either the pore-forming region or an intermediary molecule to form a complex that results in channel closure (Hervé and Sarrouilhe 2002, PMID 12566217; Pogoda et al. 2016, PMID 27229925). Interestingly, these intramolecular loop/tail interactions differentially regulate Cx43 HCs and GJs. Binding of the CT to the CL causes inhibition of Cx43 GJs, whereas it facilitates the opening of Cx43 HCs in response to known chemical stimuli (Ponsaerts et al. 2010, PMID 206343352). The opposite regulation of Cx43 HCs vs. GJs was clarified in the revised manuscript in the introduction in lines 76-80 and lines 497-499.

  1. L139 vs. L167

L139 vs. 167

L139 - In contrast, decreased Cx43 mRNA and unaltered protein levels were also reported in hippocampal tissue from patients with a complex partial seizure disorder [71]. 

L167 - Moreover, TLE is often associated with massive gliosis and activated astrocytes, which also changes Cx43 expression and function [60].                

We understand the remark of the reviewer as this is one of the main conclusions of the review, i.e. that a lot of discrepancies in Cx43 (mRNA/protein) expression can be found in literature. Possible reasons for these conflicting results in both animal models and human epilepsies are described in lines 158-160 (human data) and lines 194-198 (animal data) of the revised manuscript. However, such variations are not only restricted to epilepsy, but are often present in multiple Cx research domains, such as liver diseases.

Point 5
Please explain R0/P1/P2 Cx43 and their function (L257).

Response 5    
The phosphorylation states and their function were described in the revised manuscript in lines 199-205.

Point 6
Please move some sentences regarding the "tonabersat" from conclusion section.

Response 6    
Sentences regarding tonabersat were removed from the conclusion section in the revised manuscript.

Round 2

Reviewer 2 Report

The authors respond almost of my comments except for one important issue.

I partially agree with (can understand) statement of authors. Based on the authors' opinion, I evaluate the word 'anti-seizure drug' is not full meaning and composed of various  inconsistencies.

I should ask authors, what is the major target of this excellent review article the mechanisms of anti-epileptogenesis, anti-ictogenesis, anti-convulsive or anti-epileptic seizure actions? 

I understand these above words and anti-seizure drug cannot explain fully the author's meaning. Therefore, I must not accept the comment of author or 'anti-seizure drug'.

The author downplays terminology, and the author does not understand that it leads to be a fatal flaw in this review.

Author Response

Dear reviewer,
We understand that the word “anti-seizure drug (ASD)” does not cover the full meaning and agree to change the naming of the marketed treatments for epilepsy as “anti-epileptic drugs (AEDs)”. As these terms are often used interchangeably, it was never our intention to downplay terminology. To the best of our knowledge, we corrected terminology throughout the manuscript as suggested in the first review round. However, if this adjustment now would still not meet your expectation, we kindly ask if it would be possible to be a bit more specific on which terminology is currently seen as a flaw in the manuscript. Indeed, this manuscript covers several mechanisms involved in the field of epilepsy, in particular anti-ictogenesis, anti-convulsive and anti-epileptic actions. In addition to changing the terminology of "ASD" to “AED”, we would therefore like to propose a new, broader title, being: “Astrocytic connexin43 channels as candidate targets for epilepsy treatment”. We hope that these adaptations improve the terminology in the manuscript, but remain open for additional suggestions by the reviewer.